# A Review and a Framework of Variables for Defining and Characterizing Tinnitus Subphenotypes

**DOI:** 10.3390/brainsci10120938

**Published:** 2020-12-04

**Authors:** Eleni Genitsaridi, Derek J. Hoare, Theodore Kypraios, Deborah A. Hall

**Affiliations:** 1Hearing Sciences, Division of Clinical Neuroscience, School of Medicine, University of Nottingham, Nottingham NG7 2RD, UK; derek.hoare@nottingham.ac.uk (D.J.H.); deborah.hall@nottingham.ac.uk (D.A.H.); 2National Institute for Health Research Nottingham Biomedical Research Centre, Nottingham NG1 5DU, UK; 3School of Mathematical Sciences, University of Nottingham, Nottingham NG7 2RD, UK; theodore.kypraios@nottingham.ac.uk; 4Queens Medical Centre, Nottingham University Hospitals NHS Trust, Nottingham NG7 2UH, UK; 5University of Nottingham Malaysia, Semenyih 43500, Selangor Darul Ehsan, Malaysia

**Keywords:** heterogeneity, classification, subtyping, phenotype, unsupervised machine learning, cluster analysis, latent class analysis

## Abstract

Tinnitus patients can present with various characteristics, such as those related to the tinnitus perception, symptom severity, and pattern of comorbidities. It is speculated that this phenotypic heterogeneity is associated with differences in the underlying pathophysiology and personal reaction to the condition. However, there is as yet no established protocol for tinnitus profiling or subtyping, hindering progress in treatment development. This review summarizes data on variables that have been used in studies investigating phenotypic differences in subgroups of tinnitus, including variables used to both define and compare subgroups. A PubMed search led to the identification of 64 eligible articles. In most studies, variables for subgrouping were chosen by the researchers (hypothesis-driven approach). Other approaches included application of unsupervised machine-learning techniques for the definition of subgroups (data-driven), and subgroup definition based on the response to a tinnitus treatment (treatment response). A framework of 94 variable concepts was created to summarize variables used across all studies. Frequency statistics for the use of each variable concept are presented, demonstrating those most and least commonly assessed. This review highlights the high dimensionality of tinnitus heterogeneity. The framework of variables can contribute to the design of future studies, helping to decide on tinnitus assessment and subgrouping.

## 1. Introduction

Tinnitus, an involuntary perception of sound in the absence of an external acoustic stimulus, is a very common condition [1]. Its impact on the affected individual can range from minimal to severe [2]. To date there is no cure for the majority of tinnitus cases. A popular speculation is that pathophysiological heterogeneity of tinnitus hinders the development of effective treatments [3]. There are at least two mechanistically distinct types of tinnitus: objective tinnitus (perception caused by an acoustic source within the patient’s body) and subjective tinnitus (phantom perception). Pathophysiological heterogeneity is speculated for the much more common type, i.e., subjective tinnitus. It is well known that various pathologies, such as different pathologies causing hearing loss, are risk factors for subjective tinnitus [4,5]. However, the mechanisms up to the point of tinnitus emergence and those related to differences in tinnitus characteristics remain unclear, and there is no established objective way to investigate them. Another way to examine tinnitus heterogeneity is in terms of differences in phenotypic characteristics. Understanding phenotypic patterns in tinnitus patients could provide an essential basis for the investigation of tinnitus mechanisms and treatments [3].

The problem of heterogeneity is common to numerous medical conditions, such as schizophrenia, diabetes, and asthma to name but a few [6,7,8]. Sometimes heterogeneity is better explained as a continuous spectrum of variability (dimensional), while other times it can be explained by defining distinct subgroups (categorical, taxonomic) [9]. Different terms can be used for subgroups of patients with a medical condition, depending on the characteristics and the purpose of subgrouping. We use the term subtypes when the intention is to guide treatment decisions, and the term endotypes when it is established that the subgroups are associated with different underlying mechanisms [8,10]. We also use the term (sub)phenotypes when subgrouping is based on differences only in observable characteristics [8].

Identifying tinnitus subtypes is a longstanding objective in tinnitus research and multiple elements have been suggested for subtyping tinnitus including the presence of auditory disorders, psychopathological disorders, and somatosensory–auditory interactions [11]. However, the task of tinnitus subtyping is far from trivial. The lack of objective measures for the perception [12] and lack of standardized assessment more generally [13] hinders such approaches. In addition, there are widespread individual differences in characteristics such as onset (e.g., conditions related to tinnitus onset and age at tinnitus onset), sound qualities (e.g., pitch and loudness), impact (e.g., degree of symptom severity), and other individual factors (e.g., presence of other comorbidities). This makes it difficult to properly assess all relevant aspects in one study.

A major obstacle to investigating subgroups is that there is no established or standardized methodology. One commonly used hypothesis-driven method is to subgroup the population based on one or more characteristics according to the opinion of the researcher, and subsequently to investigate the characteristics of the resulting subgroups. Alternative data-driven methods have been used extensively in other medical fields to define subgroups, but less so for tinnitus. These include unsupervised learning techniques such as cluster analysis (clustering) and latent class analysis [6,14,15,16]. Although such techniques have existed for many decades, methodological advances have been achieved through computer and mathematical sciences [17,18,19]. These techniques always find a subgrouping solution even if there is no actual structure in the data and so validation of results is a challenge. Mathematical validation techniques can be used to assess the compactness, connectedness, separation, and stability of identified clusters. For example, the silhouette width is a combination measure for intra-cluster compactness and inter-cluster separation [20]. Similar to hypothesis-driven approaches, characteristics of identified subgroups can subsequently be investigated. Finally, another approach would be to define subgroups based on their response to a specific treatment, and to then investigate whether those subgroups have distinct phenotypic characteristics. Irrespective of the approach, a lack of standardized methodology means that various statistical techniques are employed. For example, both bivariate statistical tests and multivariable classification algorithms have been used for assessing subgroup characteristics see for example [21].

This review summarises the experimental studies that investigated subgroups of tinnitus with distinct characteristics. The main objective of this work was to create a descriptive framework of variables that have been used for defining and characterising tinnitus subphenotypes. The purpose of this framework is to serve as a reference for the design of future research studies investigating tinnitus mechanisms or treatments. A secondary objective was to summarise and evaluate methodological aspects and findings of the data-driven studies that we identified. Our findings highlight the high dimensionality of tinnitus heterogeneity, identify gaps in current literature, and provide a basis for the design of future studies.

## 2. Materials and Methods

### 2.1. Search Strategy and Screening Process

Eligible for inclusion were all original research articles investigating differences between tinnitus subgroups relevant to characterizing tinnitus subphenotypes. Although some tinnitus profiles might be better defined along a continuum, for the purposes of this review we included only studies that defined mutually exclusive tinnitus subgroups. Animal studies, case-reports, reviews, study protocols, editorials, erratum, letters to the editor, and articles not in the English language were excluded. Case reports and studies with small sample sizes (less than 30 participants with tinnitus) were also excluded.

PubMed was searched on 15 August 2020 using the following search string: (tinnitus(Title)) AND (profil*(Title/Abstract) OR subtyp*(Title/Abstract) OR subgroup*(Title/Abstract) OR class*(Title/Abstract) OR subphenotyp*(Title/Abstract) OR phenotyp*(Title/Abstract) OR type*(Title) OR group*(Title) OR cluster*(Title/Abstract) OR unsupervised(Title/Abstract)). An initial list of six articles meeting our inclusion criteria was identified by this search methodology [22,23,24,25,26,27].

The screening process involved three stages. In the first stage, one reviewer screened all titles and abstracts and rated each as follows: 1 = no relevant data in the abstract or a non-eligible study type, 2 = abstract reported subgroup comparisons relevant to characterizing tinnitus subphenotypes but lacked certain key details, and 3 = abstract clearly reported such comparisons. Rating 1 included reports of analyses investigating the association of two variables measuring the same characteristic, or of variables not contributing to tinnitus phenotyping (e.g., association of participation in self-help groups with tinnitus knowledge). Rating 2 included reports of analyses without statistical testing. Articles rated as 1 were excluded. In the second stage, for articles rated as 2 or 3, abstracts were reviewed again to rate whether characterizing tinnitus subphenotypes was a main, secondary, or exploratory aim of the article as follows: Main = a relevant tinnitus subgroup comparison was a main aim or focus; Secondary = a relevant tinnitus subgroup comparison was a secondary aim; Exploratory = a relevant tinnitus subgroup comparison was mentioned as a post-hoc exploratory step. To reduce individual bias, this stage was conducted independently by two reviewers and discrepancies were discussed and resolved. Articles rated as exploratory were excluded. In the third stage, for the remaining articles, one reviewer conducted full text screening and data extraction using an excel template which reported each variable in a separate row.

### 2.2. Data Extraction and Synthesis

Extracted information included general study information (e.g., title, authors, year, and publisher) and methodological information (e.g., sample size and type of analysis). Data were extracted only for those analyses based on data from at least 30 participants, and only for analyses related to main or secondary aims. Information from analyses that did not report clear comparisons among groups was not extracted. For relevant analysis (could be more than one per study) data were extracted about how each included variable was measured and whether it was used for defining or for comparing the subgroups. In the latter case, information about whether the variable differed significantly across groups or was important for classification was also noted. Statistical significance was based on the definition chosen by the authors (as long as the significance level (alpha) was 0.05 or smaller), which differed across studies. For example, some studies corrected *p*-values for multiple comparisons [22,23], while others did not [25,28]. In the case of application of supervised learning algorithms for classification, the importance of certain variables was dependent on researcher definitions or cut-offs. For example, variables were noted as important if they were included in the best model during a feature selection process [29], if they were in the top 5 or 50% of variable importance [28,30], or if they had non-zero coefficients in Least Absolute Shrinkage and Selection Operator (LASSO) regression [31]. Based on their design, analyses were classified into one of three categories:
Hypothesis-driven: researchers predefined tinnitus subgroups based on one or more variables (hypothesizing that the chosen variables can define phenotypically distinct subgroups) and subsequently compared them against other characteristics. Studies assessing a treatment outcome that compared treatment response across predefined tinnitus subgroups, were also classified as hypothesis-driven.Data-driven: analyses used unsupervised learning for defining tinnitus subgroups. The details of the statistical methods used and the identified subgroups were also extracted and charted.Treatment response: analyses based tinnitus subgroup definition on the response to a tinnitus treatment (e.g., responders versus non-responders).

From the data extracted, a conceptual framework for categorizing variables was created. Variables were initially categorized into a main domain: either tinnitus-specific (i.e., traits that can be measured only in people with tinnitus) or non tinnitus-specific (i.e., traits that can be measured in both people with and without tinnitus). Then, variables were categorized into subdomains (e.g., mental health and tinnitus perceptual characteristics). Finally, each variable concept was labelled. This was an iterative process involving discussion between authors. Various sources were consulted for deciding on the labels including literature related to the development or use of the identified measures and the ICD-10 classification of clinical disorders [32]. Some labels were based on headings used during the development of the European School for Interdisciplinary Tinnitus Research Screening Questionnaire (ESIT-SQ) [33] while others, particularly the ‘impact and reactions’ subdomain, were based on tinnitus domains grouped according to the World Health Organization [34]. To further reduce personal biases of the coder, the final framework was reviewed by all authors and changes were made according to mutual consensus. The same label was given to variables measuring the same concept, even if the exact assessment methodology differed.

The extracted data were analysed using R version 4.0.2 [35] to create a variable framework and calculate frequency statistics for each variable, including how many times it had been used for subgrouping or characterizing subgroups, and how many times it had been shown to differ significantly among tinnitus subgroups. R packages used included ggplot2 [36], collapsibleTree [37], and flextable [38].

## 3. Results

### 3.1. Included Studies: Overview and Methods

Results from the screening and selection process are shown in Figure 1. Overall, 64 articles were eligible for inclusion spanning publication years from 1989 to 2020 (Figure 2). The histogram demonstrates a growing activity in tinnitus phenotyping, particularly over the past 5 years, and most recently in attempts to characterize subgroups based on treatment response.

Fifty-five articles reported hypothesis-driven analyses [22,23,24,25,28,29,30,31,39,40,41,42,43,44,45,46,47,48,49,50,51,52,53,54,55,56,57,58,59,60,61,62,63,64,65,66,67,68,69,70,71,72,73,74,75,76,77,78,79,80,81,82,83,84,85], eight articles reported data-driven analyses [27,50,86,87,88,89,90,91], and two articles reporting treatment response analyses (responses to cochlear implantation and physiotherapy) [92,93]. One article reported both hypothesis-driven and data-driven analyses [50]. The use of data-driven methods by applying unsupervised learning seems to have had rather patchy uptake as some years have no publications reporting this method (Figure 2).

Most of the included articles (50/64) used only bivariate tests to compare subgroups against other variables. Multivariable techniques used across studies included regression models (10 studies) [28,30,31,42,54,69,70,80,83,93], gradient boosted trees (1 study) [29], discriminant analysis (2 studies) [40,77], analysis of covariance (1 study) [40], analysis of variance (1 study) [66].

Table 1 summarizes the characteristics of the eight studies reporting data-driven analyses. No strong patterns emerged, perhaps due to the different study designs and different sets of variables included. Only one study used a data-driven technique (Principal Component Analysis) for variable selection [91], whereas the rest depended solely on researcher decisions. The techniques used for subgrouping included k-means and hierarchical cluster analysis, and latent class analysis. Most studies validated their results by comparing subgroups against phenotypic characteristics that had not been used for subgrouping [27,50,86,87,88,89]. One study applied a silhouette measure to mathematically validate the solution [91]. No other mathematical validation measures were reported and no study reported stability measures of the identified groups.

### 3.2. Conceptual Framework for Categorizing Variables

Data were extracted for 2559 variables reported in 132 analyses from 64 studies. The majority of these variables (80.1%; 2050/2559) were self-reported measures (10 not specified). Self-reported measures were coded as such for information traditionally assessed by self-report, such as education, income, and clinical history taking, even when the measurement method was not explicitly mentioned. The remainder were investigator-administered tests (19.5%, 499/2559), with just 9.5% (243/2559) being objective measures. One study measured carotid artery intima-media thickness and various blood components [83]. Other objective measures were for hearing function (middle ear muscle reflex, auditory brainstem responses and otoacoustic emissions) (4 studies; [51,56,80,93]) and brain anatomy or function (electroencephalography, magnetic resonance imaging, positron-emission tomography) (9 studies; [46,48,64,73,74,76,77,82,89]). Genetic data (single nucleotide polymorphisms) were only assessed in 2 studies [71,84].

Overall, there were 94 distinct variable concepts which were grouped into 15 subdomains. Eight of these were non tinnitus-specific (Figure 3) and seven were tinnitus-specific (Figure 4). In these figures, ordering of variables is somewhat arbitrary, but tries to keep conceptually similar variables close to each other. Non tinnitus-specific subdomains included demographic and socio-economic characteristics, lifestyle and exposures (e.g., noise exposure and substance use), ear and hearing function (e.g., hearing ability and presence of vertigo), mental health (e.g., symptomatology of depression or anxiety), a mixed subdomain for other clinically relevant symptoms and conditions (e.g., quality of life assessment and various comorbidities), general (non tinnitus-specific) healthcare and treatments (e.g., psychiatric treatments), brain anatomy and function assessment (e.g., structural features), and genetic profile (single nucleotide polymorphisms). Tinnitus-specific subdomains included onset related characteristics (e.g., duration), perceptual characteristics (e.g., pitch and loudness), modulating factors (e.g., effect of sound or somatic manoeuvres), associations with other conditions (e.g., between tinnitus characteristics and the hearing profile), impact and reactions to tinnitus (including many of the difficulties asked about in tinnitus-specific quality of life questionnaires), and factors related to treatments for tinnitus (e.g., number of clinician consultations and response to treatments). A specific subdomain for clinically defined tinnitus subtypes was also created, including subtypes based on likely clinical aetiology, on the presence or not of somatosensory tinnitus, and on the differentiation between typewriter and middle ear myoclonus tinnitus.

No single study considered all 94 possible factors at the same time. Nevertheless some variables were more popular choices. Figure 5 shows 61 variables that were used for subgrouping or were shown to differ significantly among subgroups or be important for classification models (referred to as being important for subphenotyping) in two or more studies. Overall tinnitus severity was both the most popular variable and the one most often shown to be important for subphenotyping. Age, hearing ability and depressive symptomatology were also often shown to be important for subphenotyping.

The more interested reader is directed to Appendix A (non tinnitus-specific variables) and Appendix A (tinnitus-specific variables) which present the variables in subdomains ordered according to frequency as well as quantitative data. The quantitative data report how many studies included each variable in their analyses, how many times (number of studies) the variable was used to define tinnitus subgroups and how many times (number of studies) it was found to differ significantly across subgroups or be important for the classification. The whole extraction table is also provided as Appendix A.

## 4. Discussion

### 4.1. Overview of Results

To the best of our knowledge this is the first review of studies investigating tinnitus subphenotypes, summarizing the increasing efforts towards tinnitus subtyping. The numerous assessed variables demonstrate the high dimensionality of tinnitus heterogeneity, but by synthesizing data across studies, a framework of variables that have been used for tinnitus subphenotyping was created. This can help identify trends and gaps in current literature, and serve as a reference to guide future studies. In addition, our review identified a limited application of unsupervised learning techniques in tinnitus literature, both in terms of number of studies and technical details. The potential of such techniques has been proven in other medical fields and should be harvested to advance tinnitus research.

### 4.2. Variables for Tinnitus Phenotyping

An attempt to order variables based on how often they have been highlighted as important for tinnitus subphenotyping was presented in Figure 5. Variables were ordered based on how many of the included studies used these for subgrouping, or showed that they differed significantly among subgroups or were important for classification models. This can help researchers decide on which variables to include in future tinnitus studies. The top four variables identified with this approach were the overall tinnitus severity, hearing ability, age, and depressive symptoms. Another way of variable ordering could be according to the ratio of the number of studies highlighting a variable as important relative to the total number of studies that assessed it. In this case, quality of life, a variable related to the concept of overall tinnitus severity, could also be recommended. However, despite providing a useful indication of variable importance, we caution that our ordering is still somewhat arbitrary and choice of variables reflects inherent biases within the research community.

Although numerous variables have been considered for tinnitus subphenotyping, many of these were considered by just few studies. Few tinnitus studies evaluated objective markers such as hearing function, neuroimaging, genetic and biochemical markers. Objective measures have been used much more extensively in other fields such as for data-driven depression subtyping [15], and should be considered more often in future tinnitus studies in addition to the other relevant phenotyping variables.

The high dimensionality of variable concepts and measures that are being used for tinnitus phenotyping emphasizes the need for a consensus for standardized tinnitus assessment. This would be extremely beneficial, improving design of future studies and allowing comparisons of results across studies. Despite ongoing efforts for such standardization [33,94], identifying the subsets of variables that are essential and sufficient to characterize tinnitus subtypes is a challenging task that will require coordinated effort across the community and large-scale data analysis.

### 4.3. Methodological Considerations

Regarding analysis methods, the majority of identified studies used a hypothesis-driven approach to define subgroups. The advantage with this design is that if the resulting subgroups have distinct characteristics the researcher’s original hypothesis is validated. Nevertheless, such studies use only one or (at best) just a few variables to define subgroups, while it is far more likely that a diverse set of variables is required to define homogeneous tinnitus subphenotypes.

Data-driven studies were less common and their methods and results were very heterogeneous. Such studies can be of significant importance in advancing understanding for tinnitus heterogeneity, since they can consider multiple variables simultaneously. Results, however, should be interpreted with caution since they depend considerably on the details of the method used, including choice of the algorithm and its parameters, and the selection of included variables [95]. Therefore the resulting subgroups might not be clinically relevant. This underlines the importance of validating the results of such data-driven approaches, which is a step that is frequently overlooked. For example, assessment of cluster stability was not assessed in any of the studies included in this review, and should be considered in future research. In addition to mathematical validation, efforts to replicate findings in new datasets would be very important for establishing any identified patterns of tinnitus subphenotypes.

It is worth noting that there is at least one additional eligible study using clustering analysis for tinnitus phenotyping but this was published after our data extraction process was completed and, therefore, is not included [96]. In this study, X-means clustering was applied on self-report data from 1228 participants. Four subgroups of tinnitus were identified and novel visualisation techniques were implemented for graphical overview of their characteristics. Clustering validation was not addressed in this study. No new variable concepts were assessed in this study, and thus including it in our review would not have changed the variables framework (Figure 3 and Figure 4). Clustering analysis was also applied by Niemann, Boecking [29] in a so-called supervised clustering way [97]. Specifically, researchers initially calculated a value for each variable per participant representing the variable’s importance for the classification into tinnitus symptom severity subgroups. Subsequently, they applied hierarchical clustering on these values to identify subgroups of patients with similar importance values. Characteristics of the resulting subgroups were presented in a descriptive way.

Our search also identified two studies that defined tinnitus subgroups based on the response to a specific treatment [92,93]. Such studies are very important since they directly link phenotypes to differential treatment response, suggesting treatment predictors. However, results are specific to the investigated treatments. Another way to link phenotypes to treatment response is by comparing treatment response of predefined subgroups in a hypothesis-driven design [67]. Such analysis should better be justified and prespecified in a study protocol in order to increase the strength of evidence [98,99]. It should be noted that our search was not optimized to identify phenotypes linked to specific treatment responses. What is more, treatment response is often measured with numerical variables, and for this review if a study did not report comparisons between distinct subgroups it was not eligible for inclusion. This led, for example, to the exclusion of a recent study exploring the effect of many variables on the outcome of numerous treatments for tinnitus [100].

The tinnitus field has a lot to learn from the designs of studies aiming to subtype other clinical conditions. For example, longitudinal studies investigating distinct trajectories of the symptomatology, which are common in other fields such as in schizophrenia research [6], would provide novel insights into tinnitus heterogeneity. As far as statistical methods are concerned, although choosing and implementing the most appropriate analysis protocol is not always an easy task, it should be given great consideration. Due to the increased need for the investigation of heterogeneity and subtypes for numerous medical conditions, the development of guidelines for conducting such studies and assessing their quality would be very beneficial. Such guides would increase the value of reviews and meta-analysis and would provide standards for future studies.

Reviews summarizing current practice and common mistakes to be considered in future research are also very beneficial. For data-driven studies there are numerous such reviews from other fields [16,17,95]. For example, Horne, Tibble [95] reviewed the application of cluster analyses for asthma subtyping and highlighted issues related to clustering multimodal data, the clustering process itself, and documentation of the process. Finally, considering that novel statistical and computational methods, both for the analysis per se and for the validation of results, are constantly being developed e.g., [101], interdisciplinary collaborations can help ensure their optimal application towards a better understanding of heterogeneity of medical conditions.

### 4.4. Limitations

Our review does have limitations. Although the search was comprehensive for PubMed, systematic reviews should consider searching other electronic databases such as SCOPUS and Web of Science. It is also possible that articles were missed because certain search terms were not included in the syntax, although we did try to construct the search syntax to be as broad ranging as possible. In addition, we chose to restrict the review to studies that had a main focus on investigating distinct subgroups, and excluded studies looking at tinnitus heterogeneity from a dimensional or an exploratory perspective. Including such studies would provide a more comprehensive review of the literature on tinnitus heterogeneity, but would have a different purpose and would require much greater effort and a different study design.

## 5. Conclusions

Our review has highlighted the increasing interest in tinnitus subphenotyping. In addition to summarising the existing literature, we provide a framework of subtyping variables and highlight which variables have been most popular and most successful so far in defining subphenotypes of tinnitus. In order to advance our understanding of tinnitus heterogeneity future studies should be designed based on existing knowledge and utilize novel data analysis techniques that can handle multi-dimensional data.

## Figures and Tables

**Figure 1 brainsci-10-00938-f001:**
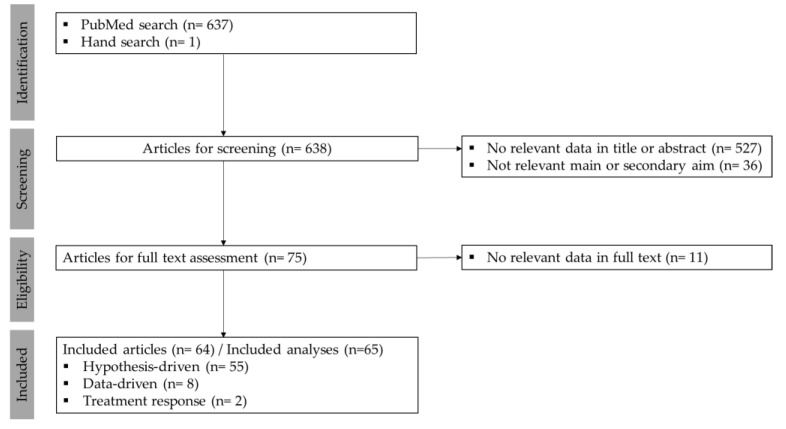
Preferred Reporting Items for Systematic Reviews and Meta-Analyses (PRISMA) flow diagram for screening and selection process.

**Figure 2 brainsci-10-00938-f002:**
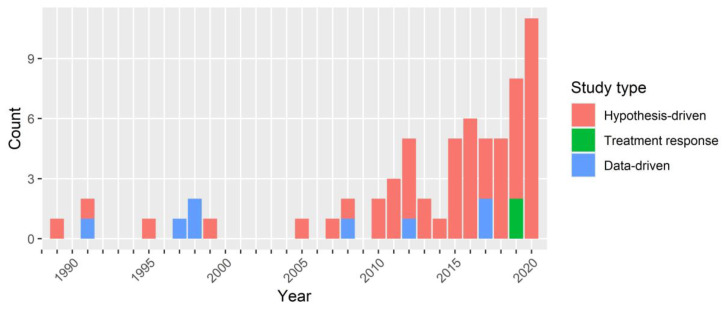
Publication years of included analyses.

**Figure 3 brainsci-10-00938-f003:**
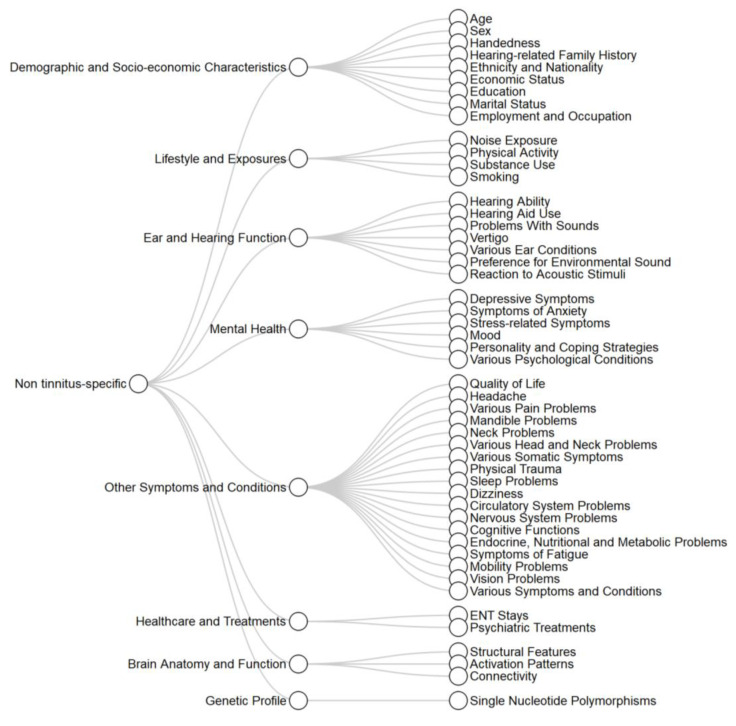
Non tinnitus-specific variables.

**Figure 4 brainsci-10-00938-f004:**
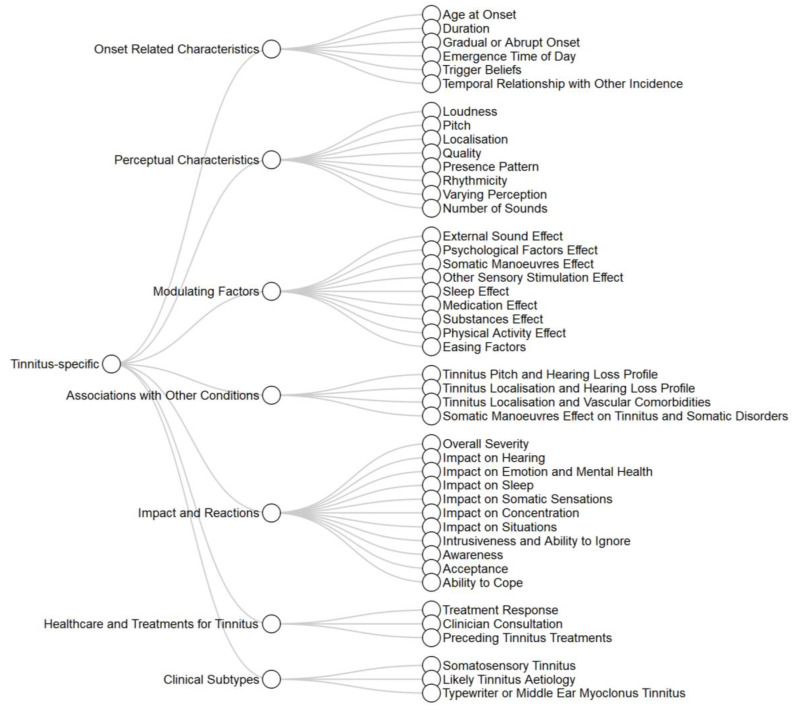
Tinnitus-specific variables.

**Figure 5 brainsci-10-00938-f005:**
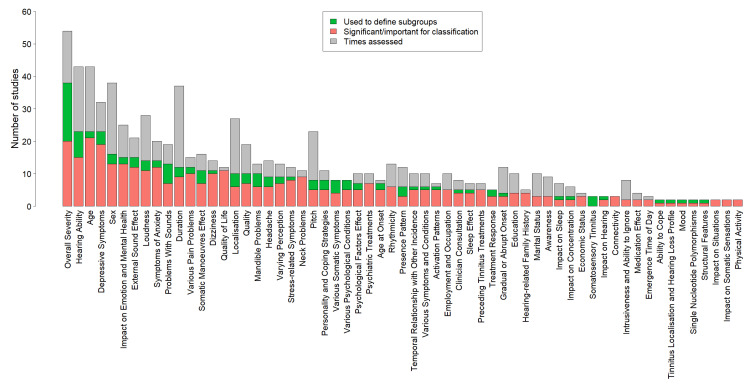
Variables that have been highlighted as important for subphenotyping (used for subgrouping, significantly differing among subgroups, or being important for classification models) in two or more studies. Green shows the number of studies that used each variable to define subgroups. Orange shows the number of studies that found this variable to differ significantly among subgroups or to be important for classification models. Grey shows the number of studies that assessed each variable but did not show it to be important for subphenotyping.

**Table 1 brainsci-10-00938-t001:** Overview of studies reporting data-driven analysis for tinnitus subphenotyping.

Studies	Variables for Subgrouping	N	Method for Subgrouping (Method for Selection of Number of Groups)	Validation	Group Descriptions
Erlandsson, Rubinstein [50]	Mood characteristics	42	Cluster analysis with k-means—distance measure not specified (prespecified three groups)	Comparison of subgroups against variables not used for subgrouping (hearing ability, age, anxiety, various tinnitus characteristics)	**3 groups:** 1. Low mood, higher tinnitus impact, anxiety, and hearing asymmetry. 2. Moderate mood. 3. High mood.
Newman, Wharton [87]	Personality and somatic symptoms (self-focused attention, somatic attention)	51	Cluster analysis minimizing squared distances from cluster means (selection among 4 groupings based on clinical interpretation)	Comparison of subgroups against variables not used for subgrouping (tinnitus impact, depressive symptomatology, tinnitus psychoacoustic characteristics)	**2 groups:** 1. Lower score on both self-attention and somatic attention measures. 2. More internally directed, higher score on the attention measures. On average more depressed, with greater emotional distress due to tinnitus, and with greater perceived tinnitus handicap.
Andersson and McKenna [86]	Depressive symptoms, hearing ability (PTA), tinnitus loudness, external sound effect on tinnitus (MML)	30	Hierarchical cluster analysis with Ward’s method and Euclidean distances (not specified)	Comparison of subgroups against variables not used for subgrouping (age, tinnitus duration and severity)	**3 groups:** 1. Low depression, average loudness, slightly above average MML and PTA. 2. High value only in depression, the youngest. 3. High values in all variables, the oldest.
Rizzardo, Savastano [88]	Personality and other mental health characteristics including symptoms of anxiety and depression	84	Hierarchical cluster analysis with complete linkage—distance measure not specified (not specified)	Comparison of subgroups against variables not used for subgrouping (demographics, clinical characteristics)	**3 groups:** 1. Higher scores for depression, anxiety, neuroticism, and hypochondriasis, more often somatic symptoms and psychotropic drug use, less ENT stays, higher tinnitus distress, and lesser degree of denial. 2. Normal psychological tests apart from marked denial. 3. One patient.
Tyler, Coelho [90]	Age, hearing function, symptoms of anxiety, depression, and stress, somatic symptoms, tinnitus duration, perceptual characteristics, modulating factors, and severity (not all variables specified)	153	2-step cluster analysis (SPSS) (tested solutions with 4–6 clusters; chose 4 because it resulted in about equal group sizes)	None	**4 groups:** 1. Loud, persistent and distressing tinnitus, highest anxiety and depression scores, with loudness hyperacusis. 2. Varying tinnitus pitch and loudness, worse in noise. 3. Copers, tinnitus not influenced by touch. 4. Copers, tinnitus worse in quiet and better in noise.
Schecklmann, Lehner [89]	Three analyses using different variables: A. Tinnitus severity, duration, and localisation B. Grey matter volume (MRI) C. Brain glucose metabolism (PET)	44	Hierarchical cluster analysis with Ward’s method and Euclidean distances (forced to 2 groups in order to have sample sizes with sufficient statistical power)	Comparison of subgroups against variables not used for subgrouping (age, sex, hearing function, and tinnitus duration, localisation and severity)	**2 groups from each of the three analyses:** A. Unilateral versus mainly bilateral tinnitus. B. Higher versus lower grey matter volume in medial superior prefrontal, cingulate, temporal, insular, orbital frontal, temporal, pre- and post-central and thalamic areas. C. Higher versus lower glucose metabolism in middle and superior temporal, precuneus, and superior parietal areas.
van den Berge, Free [91]	Two analyses using different variables: A. The variables with the highest loading on each of 8 components from a principal components analysis on 30 variables (depression or anxiety symptoms, hearing ability, tinnitus onset-related and perceptual characteristics, modulating factors, and impact) B. 11 selected variables: Sex, hearing function, tinnitus perceptual characteristics, modulating factors, and severity	A. 976, B. 761	2-step cluster analysis (SPSS) (probably silhouette measure for selection of number of groups)	Silhouette measure	**A. 4 groups** 1. Tinnitus not easily influenced, higher hearing asymmetry, lower depression score. 2. Gradual onset of tinnitus, easily negatively influenced by loud sounds and sleep deprivation. 3. Tinnitus less loud with loud sounds, no effect from sleep deprivation or nap. 4. Acute onset tinnitus, tinnitus easily negatively influenced by loud sounds or sleep deprivation. **B. 3 groups** 1. Tinnitus not easily influenced, preference for noisy environments, tinnitus mostly unilateral, low tinnitus severity scores, lower depression scores, sounds not often uncomfortably loud. 2. Mainly males, tinnitus worse by stress, loud sounds and movement of head and neck, preference for noisy environments, sometimes sounds are uncomfortably loud, most with no or slight HL, bilateral tinnitus with variable loudness. 3. Tinnitus worse by loud sounds and stress, prefer silent environment, often find sounds uncomfortably loud, tinnitus often bilateral, most with no, slight, or asymmetric HL, variable loudness.
Langguth, Landgrebe [27]	Hearing function at 7 frequencies (0.125, 0.250, 0.5, 1, 2, 4, and 8 kHz) for each ear (categorically coded as normal, mild/moderate loss, severe/profound loss, or no data)	2838	Latent class analysis (tested 2–12 groups and used BIC to determine optimal solution)	Comparison of subgroups against variables not used for subgrouping (demographics, tinnitus severity, depressive symptomatology, and various tinnitus characteristics)	**8 groups:** 1. Lacking audiometry: phenotypic characteristics similar to average. 2. Bilateral high frequency HL: more males, more often tonal tinnitus. 3. Near normal hearing: the youngest, youngest age at onset, lowest tinnitus severity, more often tonal tinnitus. 4. Bilateral medium-high frequency (2 kHz and above) HL: more males, gradual tinnitus onset. 5. Severe pantonal HL: the oldest, more females, older age at onset, highest tinnitus severity, highest depression scores, gradual tinnitus onset. 6. Left-sided pantonal medium HL: more females, older age at onset, more often abrupt tinnitus onset, left-sided tinnitus. 7. Right-sided pantonal severe HL: more often abrupt tinnitus onset and right-sided tinnitus, less often pulsatile tinnitus. 8. Left-sided pantonal severe HL: more often left sided and constant tinnitus.

BIC: Bayesian Information Criterion; ENT: Ear, Nose, Throat; HF: High frequency; HL: Hearing Loss; MML: Minimal Masking Level; MRI: Magnetic Resonance Imaging; PET: Positron Emission Tomography; PTA: Pure Tone Average; SPSS: Statistical Package for the Social Sciences.

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
