# Peer review of "A Review and a Framework of Variables for Defining and Characterizing Tinnitus Subphenotypes"

_brainsci, 2020, doi:10.3390/brainsci10120938_

Round 1

Reviewer 1 Report

The research synthesis is comprehensive and relevant. I serve on student research project committees where we examine similar studies, so I am familiar with the format. 

This synthesis appears to be comprehensive and accurate. My only suggestion relates to the purpose. While it can be inferred that the authors are seeking to advance the treatment of tinnitus, this objective was not clearly stated. If I am incorrect in that assumption, this is all the more reason to clarify the intent. 

Author Response

Thank you for reviewing our manuscript. In response to your suggestion we edited the last paragraph of the introduction (starting in line 85) to clearly state the objectives of this review:

This review summarises the experimental studies that investigated subgroups of tinnitus with distinct characteristics. The main objective of this work was to create a descriptive framework of variables that have been used for defining and characterising tinnitus subphenotypes. The purpose of this framework is to serve as a reference for the design of future research studies investigating tinnitus mechanisms or treatments. A secondary objective was to summarise and evaluate methodological aspects and findings of the data-driven studies that we identified. Our findings highlight the high dimensionality of tinnitus heterogeneity, identify gaps in current literature, and provide a basis for the design of future studies.

Reviewer 2 Report

This review is comprehensive and summarizes hypothesis-driven or data-driven approaches very well.

Author Response

Thank you for taking the time to review our manuscript.

Reviewer 3 Report

This is an extensive review article of prior studies that addressed tinnitus subphenotypes variability. The review is well written, and the article provides a good reference for future studies addressing tinnitus.
My only comment is that the review does not provide a recommendation on the type of variables that subsequent tinnitus research should optimally consider. I would welcome such a recommendation.

Author Response

Thank you for reviewing our manuscript. In response to your suggestion we added a paragraph with a relevant discussion starting in line 267:

An attempt to order variables based on how often they have been highlighted as important for tinnitus subphenotyping was presented in Figure 5. Variables were ordered based on how many of the included studies used these for subgrouping, or showed that they differed significantly among subgroups or were important for classification models. This can help researchers decide on which variables to include in future tinnitus studies. The top four variables identified with this approach were the overall tinnitus severity, hearing ability, age, and depressive symptoms. Another way of variable ordering could be according to the ratio of the number of studies highlighting a variable as important relative to the total number of studies that assessed it. In this case, quality of life, a variable related to the concept of overall tinnitus severity, could also be recommended. However, despite providing a useful indication of variable importance, we caution that our ordering is still somewhat arbitrary and choice of variables reflects inherent biases within the research community.